# Persistent Threats: A Comprehensive Review of Biofilm Formation, Control, and Economic Implications in Food Processing Environments

**DOI:** 10.3390/microorganisms13081805

**Published:** 2025-08-01

**Authors:** Alexandra Ban-Cucerzan, Kálmán Imre, Adriana Morar, Adela Marcu, Ionela Hotea, Sebastian-Alexandru Popa, Răzvan-Tudor Pătrînjan, Iulia-Maria Bucur, Cristina Gașpar, Ana-Maria Plotuna, Sergiu-Constantin Ban

**Affiliations:** 1Faculty of Veterinary Medicine, University of Life Sciences “King Mihai I” from Timișoara, 300645 Timișoara, Romania; kalmanimre@usvt.ro (K.I.); adrianamorar@usvt.ro (A.M.); ionelahotea@usvt.ro (I.H.); razvan.patrinjan.fmv@usvt.ro (R.-T.P.); bucur_iulia@ymail.com (I.-M.B.); cristina.gaspar@usvt.ro (C.G.); anamaria.plotuna@usvt.ro (A.-M.P.); 2Research Institute for Biosecurity and Bioengineering, University of Life Sciences “King Mihai I” from Timișoara, 300645 Timișoara, Romania; 3Faculty of Biotechnology of Animal Resources, University of Life Sciences “King Mihai I” from Timișoara, 300645 Timișoara, Romania; adelamarcu@usvt.ro; 4Independent Researcher, 300668 Timișoara, Romania; bansergiuconstantin@gmail.com

**Keywords:** biofilms, food safety, foodborne pathogens, sanitation, food-processing environments

## Abstract

Biofilms are structured microbial communities that pose significant challenges to food safety and quality within the food-processing industry. Their formation on equipment and surfaces enables persistent contamination, microbial resistance, and recurring outbreaks of foodborne illness. This review provides a comprehensive synthesis of current knowledge on biofilm formation mechanisms, genetic regulation, and the unique behavior of multi-species biofilms. The review evaluates modern detection and monitoring technologies, including PCR, biosensors, and advanced microscopy, and compares their effectiveness in industrial contexts. Real-world outbreak data and a global economic impact analysis underscore the urgency for more effective regulatory frameworks and sanitation innovations. The findings highlight the critical need for integrated, proactive biofilm management approaches to safeguard food safety, reduce public health risks, and minimize economic losses across global food sectors.

## 1. Introduction

Biofilms are complex microbial communities that attach to surfaces and are embedded in a self-produced extracellular matrix. They present significant challenges in the food industry by contributing to contamination, food spoilage, equipment damage, and economic losses [1].

In food-processing environments, biofilms frequently develop on stainless steel, plastic, rubber, and other contact surfaces due to constant exposure to nutrients, moisture, and suboptimal cleaning [2]. These microbial communities not only serve as reservoirs for spoilage organisms but are also implicated in serious foodborne outbreaks involving pathogens such as *Listeria monocytogenes*, *Salmonella* spp., and *Escherichia coli* O157:H7 and sometimes *Campylobacter* spp. The formation of biofilms enables these pathogens to persist despite regular cleaning, posing persistent threats to food safety and shelf life [3,4,5].

The ability of bacteria to form biofilms is also closely linked to their survival under environmental stresses, including sanitizers, desiccation, and temperature fluctuations. Potentially pathogenic bacteria originating from wild game meat have also been identified as emerging contributors to contamination routes in food chains with the capacity to participate in biofilm formation and persistence within processing environments [6]. Within the biofilm, microbial cells exhibit altered phenotypes, including increased resistance to antimicrobial agents up to 1000-fold compared to their planktonic (free-floating) counterparts [7]. This resistance complicates decontamination efforts and facilitates repeated contamination cycles, leading to recurring product recalls and compromised consumer safety [8].

Importantly, biofilm formation is not limited to single-species populations; multi-species communities are more commonly found in real-world food settings. These diverse biofilms benefit from synergistic interactions, such as metabolic cooperation and shared resistance mechanisms, which enhance their structural resilience and adaptability [9]. Moreover, biofilms facilitate horizontal gene transfer, accelerating the spread of antimicrobial resistance genes (ARGs) within and across species, further complicating infection control and contributing to the global antimicrobial resistance (AMR) crisis [10,11].

From a regulatory standpoint, biofilm-associated risks are increasingly acknowledged in frameworks such as ISO 22000 and Codex Alimentarius, which emphasize proactive monitoring and stringent hygiene protocols. Despite these guidelines, the implementation of biofilm-specific detection and prevention strategies remains limited in many sectors due to cost, lack of awareness, or technological gaps [12].

Given the ubiquity and persistence of biofilms in food environments, it is critical to understand their formation dynamics, resistance mechanisms, and ecological implications. This knowledge supports the development of targeted interventions, ranging from enzymatic disruptors and bacteriophages to surface modifications and real-time monitoring systems, to reduce contamination risks and enhance food safety.

This review provides an extensive analysis of biofilm formation mechanisms, impacts on food safety and public health, detection methods, control strategies, and the economic implications across various food sectors including dairy, meat, and seafood. A special focus is placed on the global perspective of economic losses, emphasizing the urgency for advanced biofilm control strategies.

## 2. Mechanisms of Biofilm Formation in the Food Industry

The formation of a biofilm begins with the reversible attachment of free-floating microorganisms to a surface. This initial adhesion is mediated by weak physicochemical interactions, such as van der Waals forces and hydrophobic effects. If conditions are favorable, the attachment becomes irreversible, involving the production of adhesins and the beginning of extracellular polymeric substances matrix synthesis (EPS), anchoring the cells firmly to the surface [13].

Following irreversible attachment, the microorganisms begin to proliferate and form microcolonies. During this phase, the production of EPS increases, facilitating cell-to-cell adhesion and communication. The microcolonies serve as the foundation for the developing biofilm, allowing for the establishment of complex microbial communities [14]. As the microcolonies grow, the biofilm enters the maturation phase. In this stage, the biofilm develops a complex, three-dimensional structure characterized by the formation of channels and pores that allow for nutrient and waste exchange. The EPS matrix becomes more robust, providing mechanical stability and protection against environmental stresses, including antimicrobial agents [15].

The final stage of the biofilm life cycle is dispersion, where cells are released from the mature biofilm to colonize new environments. This process can be triggered by various factors, such as nutrient depletion or changes in environmental conditions. Dispersed cells can revert to the planktonic state or initiate the formation of new biofilms elsewhere [16]. EPS is a critical component of biofilms, constituting up to 90% of the biofilm’s dry mass. They are composed of a complex mixture of biopolymers, including polysaccharides, proteins, lipids, and extracellular DNA [17]. The EPS provide the structural framework of the biofilm, facilitating the adhesion of cells to surfaces and to each other. This matrix maintains the biofilm’s architecture and protects the embedded microorganisms from mechanical forces and shear stress [18].

The matrix acts as a protective barrier, shielding the microbial community from environmental threats, including desiccation, ultraviolet radiation, and antimicrobial agents. It impedes the penetration of antibiotics and disinfectants, contributing to the increased resistance of biofilm-associated microorganisms [19].

EPS facilitates the retention of nutrients within the biofilm and enables the establishment of nutrient gradients. The matrix’s porous nature allows for the diffusion of nutrients and metabolic waste, supporting the survival and growth of the microbial community [20].

### 2.1. Genetic Regulation of Biofilm Formation

Biofilm formation is a complex, multicellular behavior regulated by quorum sensing (QS), which is a mechanism that enables bacteria to coordinate gene expression based on population density. This process involves the production, release, and detection of signaling molecules known as autoinducers. Upon reaching a threshold concentration, these autoinducers bind to specific receptors, triggering transcriptional changes that facilitate biofilm development [21].

In Gram-negative bacteria, these receptors are typically LuxR-type proteins located in the cytoplasm, which bind freely diffusible acyl-homoserine lactones (AHLs) before directly regulating gene transcription. The LuxI/LuxR QS system is well characterized. For instance, *Pseudomonas aeruginosa* utilizes the Las and Rhl systems, where LasI synthesizes N-(3-oxododecanoyl)-homoserine lactone (3OC12-HSL), which binds to the LasR receptor to regulate genes involved in biofilm maturation and virulence. Similarly, the RhlI/RhlR system controls the production of rhamnolipids, contributing to biofilm architecture and dispersal [21,22].

In Gram-positive bacteria, QS signals are usually oligopeptides detected by membrane-bound histidine kinase receptors, which then activate a cytoplasmic response regulator via a two-component phosphorylation cascade. The accessory gene regulator (Agr) QS system is prominent in *Staphylococcus aureus*, and it regulates the expression of surface proteins and exoproteins, influencing biofilm development and detachment. Mutations in the *agr* locus can lead to enhanced biofilm formation due to the sustained expression of surface adhesins [23]. The LuxS/AI-2 system represents a universal QS mechanism present in both Gram-negative and Gram-positive bacteria. AI-2, synthesized by the LuxS enzyme, mediates interspecies communication and has been implicated in biofilm formation across diverse bacterial species. For example, in *Escherichia coli*, AI-2 influences the expression of genes involved in motility and adhesion, affecting biofilm dynamics. Targeting QS pathways offers a promising strategy for controlling biofilm-related issues in food-processing environments. By disrupting QS signaling, it is possible to prevent biofilm formation and reduce the risk of contamination, thereby enhancing food safety and quality [24,25]. There are several known mechanisms to disrupt the QS signaling. One approach employs the enzymatic degradation of signaling molecules: enzymes like AiiA lactonase hydrolyze the homoserine-lactone ring of N-acyl homoserine lactones (AHLs), effectively inactivating these autoinducers and preventing QS activation. AiiA, produced by *Bacillus* species, has been shown to disrupt QS in *Pseudomonas aeruginosa*, leading to reduced biofilm formation and virulence [26]. A related mechanism involves signal molecule modification, where oxidoreductases from species like *Rhodococcus* chemically alter AHL structures to prevent receptor binding [27]. A second tactic is the inhibition of signal synthesis, in which compounds such as plant-derived curcumin block the production of autoinducers by interfering with key synthase enzymes [28]. Another line of defense is signal receptor antagonism, where structural analogues of AHL molecules competitively bind to QS receptors (e.g., LuxR or LasR), thereby blocking native autoinducer signals and disrupting gene regulation [29]. Finally, downstream signaling interference targets the regulatory cascade directly, such as the RNAIII-inhibiting peptide (RIP), which blocks the Agr QS system in *Staphylococcus aureus*. RIP prevents phosphorylation of the TRAP protein, interrupting QS-dependent virulence gene expression and biofilm development [30].

### 2.2. Horizontal Gene Transfer and AMR

Horizontal gene transfer (HGT) is a pivotal mechanism by which bacteria acquire and disseminate antibiotic resistance genes (ARGs), significantly contributing to the global challenge of antimicrobial resistance (AMR). Within biofilms, the close proximity of bacterial cells and the protective extracellular matrix create an ideal environment for HGT, facilitating the exchange of genetic material through various mechanisms [31]. The primary modes of HGT include conjugation, transformation, and transduction.

Conjugation is the direct cell-to-cell transfer of plasmid DNA via sex pili, and it occurs with notable efficiency within biofilms due to the close physical proximity and stabilized contacts among bacterial cells. The biofilm’s extracellular polymeric substance matrix, along with mild fluid shear forces, helps maintain these cell-to-cell connections, enabling prolonged conjugative interactions. Natural conjugative plasmids not only enhance plasmid transfer in *Escherichia coli* but also promote biofilm formation, particularly in mixed-species communities. The study demonstrated that *E. coli* strains carrying conjugative plasmids developed thick, surface-associated biofilms on glass, which is likely because plasmid-encoded pili facilitated adherence and stable microcolony formation. Furthermore, these conditions foster repeated rounds of plasmid transfer, effectively making biofilms hot spots for horizontal gene transfer [32].

Transformation refers to the uptake of free, extracellular DNA (eDNA) from the environment by competent bacterial cells. In biofilms, this process is significantly enhanced due to the elevated concentration of eDNA derived from cell lysis, active secretion, and membrane vesicle release within the extracellular matrix [33]. Many bacteria, under conditions of nutrient limitation or high density, enter a temporarily “competent” state permitting DNA uptake. In biofilms, quorum-sensing-regulated competence pathways are activated by eDNA accumulation. Surface proteins involved in DNA binding and transport then facilitate DNA entry through pili-like structures into the periplasm or cytoplasm [34]. Once within the cell, DNA is processed, and single-stranded fragments are bound by recombination proteins (e.g., RecA) and integrated into the genome via homologous recombination. This incorporation of novel genetic elements, including antibiotic resistance genes (ARGs), often occurs at higher frequency in biofilms due to increased eDNA and longer exposure times [35]. Field studies involving *Neisseria gonorrhoeae* biofilms have visualized the dynamic movement and uptake of eDNA within colony structures, confirming that biofilm contexts facilitate high-efficiency DNA transfer and intercellular exchange [36].

Transduction represents the bacteriophage-mediated transfer of DNA and is increasingly recognized within biofilm communities as a key mechanism for horizontal gene spread, including antibiotic resistance genes (ARGs) [32]. In biofilms, temperate (lysogenic) and virulent (lytic) phages can infect resident bacteria more efficiently due to the high cell density and stability provided by the EPS matrix, which enables repeated infection cycles and prolonged phage–bacterium interaction [37]. Generalized transduction occurs during the phage lytic cycle when random fragments of degraded bacterial DNA are mistakenly packaged into phage capsids. In biofilms, the rich environment and frequent bacterial cell lysis increase the availability of DNA, boosting the chances of ARG encapsulation and transfer to new host cells during subsequent phage infection [38]. Specialized transduction involves temperate phages, where the incorrect excision of prophage DNA may incorporate adjacent bacterial genes into the phage genome. These “hybrid” phages then transfer these genes, sometimes encoding virulence factors or antibiotic resistance upon reinfection. Biofilms enhance this process due to elevated rates of prophage induction and localized propagation [39]. A recently identified mechanism, lateral transduction, occurs when phage replication initiates while still integrated into the host genome. This results in large stretches of adjacent bacterial DNA (up to 100 kb) being packaged during excision and subsequently transferred at higher frequencies in biofilm environments [40].

### 2.3. Multi-Species Biofilms

In real-world food-processing environments, biofilms are rarely composed of a single microbial species. Instead, they often consist of multiple species interacting synergistically. The phenomenon of synergy in biofilms refers to situations where the combined biofilm production of multiple microbial species exceeds the sum of biomass of what each species would produce independently. Increased biomass creates thicker, more protective barriers against cleaning agents and environmental stresses [41]. Multi-species biofilms form through complex interactions among different microbial species. These interactions can be cooperative, where species support each other’s growth and survival, or competitive, where they inhibit each other. The spatial organization within these biofilms plays a crucial role in their development and resilience. For instance, certain bacteria may occupy specific niches within the biofilm, providing structural stability and protection to others [42]. Mutualism and commensalism frequently occur within these communities. Certain lactic acid bacteria, for example, can consume oxygen and create localized anaerobic niches, thereby supporting the survival of facultative anaerobes. Meanwhile, commensal species may benefit indirectly from metabolic by-products or the shared EPS matrix without providing any reciprocal advantage [41].

Syntropy, a form of direct metabolic cooperation, is especially relevant under nutrient-limited conditions common in food-processing settings. One species may ferment complex carbohydrates into short-chain fatty acids, which are then used by another organism. Such cooperation promotes the formation of nutrient channels and fosters stable internal architectures capable of resisting shear forces and during flow stress [41].

In contrast, competitive exclusion also shapes biofilm communities. Some bacterial strains produce bacteriocins or engage in quorum-sensing interference to inhibit competitor species. These antagonistic interactions can lead to spatial segregation, result in micro-niches of heightened resistance, and contribute to uneven vulnerability against cleaning interventions [42].

The persistence of biofilms in food-processing environments enables a wide range of pathogenic microorganisms to evade routine sanitation efforts, thereby increasing the risk of foodborne illnesses. These pathogens, many of which are capable of forming resilient biofilms, pose significant health threats to consumers, particularly vulnerable populations such as the elderly, children, and immunocompromised individuals [1]. Table 1 provides an overview of key biofilm-associated pathogens.

Foodborne outbreaks linked to biofilm-forming bacteria across Europe reveal a troubling pattern of contamination events associated with a range of food products and processing settings. Table 2 compiles notable outbreaks in Europe. These documented incidents highlight the real-world consequences of biofilm persistence and the urgent need for enhanced surveillance, sanitation innovations, and regulatory enforcement.

### 2.4. Role of Environmental Factors

#### 2.4.1. Temperature

Temperature is a critical environmental factor that significantly influences biofilm formation, structure, and resistance in food-processing environments. Psychrotrophic bacteria such as *Pseudomonas putida*, *P. fluorescens*, and *P. aeruginosa* tend to form enhanced biofilms at refrigeration temperatures (6–10 °C), which is a phenomenon linked to increased adhesion, EPS production, and the upregulation of cold stress and energy metabolism genes. For instance, *Pseudomonas fluorescens* isolated from dairy environments displayed strong adhesion and metabolic activity at temperatures between 6 and 10 °C, particularly in protein-rich media, which promotes biofilm maturation. Similarly, *P. putida* forms enhanced biofilm structures at 10 °C, which is linked to the upregulation of genes associated with cold stress response and energy metabolism [54]. Conversely, *P. aeruginosa* biofilm biomass was observed to decrease at higher temperatures, with structural disruptions noted around 25 °C, suggesting that optimal biofilm formation occurs under cooler conditions. Despite this, established biofilms demonstrate considerable thermal resilience, often requiring short-term heat shock treatments ranging from 50 to 80 °C to achieve significant biomass reduction [55]. However, the efficacy of heat treatment can be variable, as regrowth may occur if parameters are suboptimal, partly due to the action of heat shock proteins such as DnaJ, which modulate extracellular DNA release and enhance initial adhesion [56].

#### 2.4.2. Humidity

High relative humidity plays a critical role in promoting biofilm formation by creating a moist environment that facilitates bacterial adhesion, initial colonization, and EPS matrix production. Elevated humidity prevents surface desiccation, enabling bacteria to maintain metabolic activity and adhere more effectively to stainless steel, plastic, or rubber surfaces commonly found in food-processing environments. Studies have shown that under conditions of 90–100% relative humidity, adhesion rates of bacteria such as *Listeria monocytogenes* and *Escherichia coli* increase significantly, enhancing the likelihood of biofilm establishment on food contact surfaces [57]. Moreover, high humidity levels enhance the resistance of established biofilms to disinfectants. For example, *Escherichia coli* O157:H7 biofilms grown under saturated humidity (100% RH) exhibit increased resistance to chlorine-based sanitizers compared to biofilms formed under drier conditions, which is likely due to denser EPS formation that impedes disinfectant penetration [58].

Additionally, in humid environments, biofilm-forming bacteria such as *Pseudomonas aeruginosa* and *Salmonella* spp. show delayed desiccation-induced stress responses, allowing prolonged survival on abiotic surfaces. This extended survival time increases the risk of persistent contamination in high-humidity zones, such as cold storage rooms or processing areas with high water activity [59]

These findings emphasize that controlling ambient humidity in processing environments is a key preventive strategy against biofilm formation. However, maintaining suboptimal humidity levels for biofilm development (<70%) is often challenging in industrial settings due to operational and product storage requirements, highlighting the need for integrated approaches combining environmental control with targeted sanitation [1].

#### 2.4.3. Nutrient Availability

Nutrient composition and availability play a decisive role in biofilm formation, architecture, and resilience. In *Salmonella enterica*, high-nutrient environments promote the formation of pellicle biofilms at the air–liquid interface, which is likely due to enhanced metabolic activity and EPS production. Conversely, under nutrient-limited conditions, *Salmonella* tends to form denser biofilms attached to the solid–liquid interface, adapting its structure for maximum surface adherence and nutrient capture [60].

Specific carbon sources also modulate biofilm development. In *Bacillus licheniformis*, glucose and maltose supplementation has been shown to significantly increase biofilm biomass and cell adhesion, which is a process linked to the carbon catabolite regulation of biofilm-associated genes [61].

Nutrient and oxygen gradients within biofilms create zones of slow-growing or dormant cells. These cells are less susceptible to antibiotics, which often target actively dividing bacteria [62]. Moreover, nutrient stress can trigger a survival response in various bacterial species, promoting biofilm formation as a protective strategy against unfavorable conditions, often leading to increased antimicrobial resistance within the biofilm [14]. Nutrient stress induces stress response pathways (e.g., rpoS-mediated responses), the upregulation of efflux pumps, and the expression of biofilm-specific resistance genes, all of which enhance antimicrobial tolerance [62].

### 2.5. Detection and Monitoring Techniques

Detecting biofilms in food-processing environments is crucial for ensuring food safety and quality. Various techniques have been developed to identify and monitor biofilm formation. Table 3 summarizes commonly used methods for the detection and characterization of biofilms in food industry settings, outlining their principles, key advantages, limitations, and representative application examples.

The early detection of biofilms is challenging due to their microscopic size and the complex environments in food-processing facilities. Traditional detection methods may not be sensitive enough to identify initial biofilm formation, leading to persistent contamination issues [63].

Fluorescence microscopy, often combined with specific staining methods like SYTO9 or propidium iodide, allows for the visualization of biofilms on surfaces. These techniques provide detailed images of biofilm structure and viability, aiding in the assessment of cleaning protocols [57].

Polymerase chain reaction (PCR) and quantitative PCR (qPCR) are employed to detect and quantify biofilm-associated genes. These methods offer high sensitivity and specificity, enabling the identification of biofilm-forming microorganisms even at low concentrations [64].

Advanced biosensors, such as electrochemical impedance spectroscopy (EIS) sensors, have been developed for the real-time monitoring of biofilm growth. These sensors detect changes in electrical properties caused by biofilm formation, providing rapid and continuous assessment [65].

**Table 3 microorganisms-13-01805-t003:** Comparative analysis of biofilm detection techniques.

Method	Principle	Advantages	Limitations	Application Example	Reference
Microscopy (e.g., CLSM, SEM)	Direct visualization of cells and matrix structure	High spatial resolution, biofilm morphology analysis	Expensive equipment, sample prep required, time consuming	Biofilm structure on stainless steel in dairy tanks	[66].
PCR (conventional or qPCR)	Detects biofilm-forming genes or microbial DNA	High sensitivity and specificity, fast	Requires DNA extraction, cannot differentiate live/dead cells	Detection of *L. monocytogenes* in food contact surfaces	[63].
Fluorescence staining	Uses dyes (e.g., SYTO9, PI) to label viable/dead cells	Differentiates between live and dead cells, visual feedback	Requires fluorescence microscopy, limited field applicability	Biofilm viability in meat slicers	[64]
ATP Bioluminescence	Measures microbial ATP as indicator of contamination	Rapid, user-friendly, widely used in industry	Cannot distinguish between planktonic and biofilm-associated cells	Hygiene audits in food production lines	[57]
Biosensors	Detects metabolites or signaling molecules (e.g., AI-2)	Real-time monitoring potential, can be integrated into production systems	Still under development, specificity and robustness vary	Online biofilm detection in water pipelines	[65]
Culture-based methods	Plate counting after surface swabbing	Simple, inexpensive, allows strain isolation	Time-consuming, misses viable but non-culturable (VBNC) cells	Routine swabbing in meat or produce facilities	[7]
Spectroscopy (e.g., FT-IR, Raman)	Analyzes chemical composition of biofilms	Non-destructive, biochemical fingerprinting	Requires specialized instruments, data interpretation complexity	Material characterization in beverage equipment	[67]

Legend: CLSM (confocal laser scanning microscopy), SEM (scanning electron microscopy), PCR (polymerase chain reaction), qPCR (quantitative polymerase chain reaction), SYTO9 (SYTO 9 fluorescent dye), PI (propidium iodide), ATP (adenosine triphosphate), AI-2 (autoinducer-2), VBNC (viable but non-culturable), FTIR (Fourier transform infrared spectroscopy), and Raman (Raman spectroscopy).

The development of real-time monitoring systems is essential for proactive biofilm management. Such systems can provide immediate feedback on biofilm presence, allowing for timely intervention and reducing the risk of product contamination. Implementing these technologies can enhance food safety protocols and minimize economic losses associated with biofilm-related issues [16].

Despite a strong theoretical foundation, there is a lack of comparative studies evaluating biofilm detection methods, such as PCR, biosensors, and microscopy, in real food-processing environments. One notable field study assessed *Listeria monocytogenes* biofilms on stainless steel and polypropylene surfaces, finding that droplet digital PCR (ddPCR) correlated closely with traditional culture counts (Pearson r = 0.864 for steel; r = 0.725 for polypropylene), demonstrating ddPCR’s practical utility in situ [68]. Conversely, conventional microscopy techniques like crystal violet assays remain predominant in many facilities despite their lower sensitivity. Emerging biosensor technologies, such as electrochemical impedance spectroscopy (EIS), have shown promise in pilot-scale trials by detecting biofilm initiation within hours and enabling timely cleaning interventions. However, their industrial deployment is still limited [69]. Another comparative analysis involving coagulase-negative staphylococci evaluated four detection methods: PCR targeting the *ica* locus, tube adherence, tissue culture plate (TCP) assay, and Congo red agar (CRA) method. Results revealed sensitivities of 82%, 82%, and 81% for PCR, tube, and TCP methods, respectively (with PCR as the reference), whereas CRA lagged behind; importantly, this study occurred under controlled lab conditions rather than industrial settings [70]. Advanced imaging techniques such as confocal laser scanning microscopy, atomic force microscopy (AFM), and optical nanosensors can provide detailed structural insights in laboratory models, but their high cost, technical complexity, and lack of scalability restrict widespread use [71].

## 3. Control and Prevention Strategies

Traditional cleaning and sanitation methods in the food industry primarily involve mechanical cleaning, chemical sanitizers, and thermal treatments. Mechanical cleaning includes manual scrubbing and the use of high-pressure water jets to remove visible debris and biofilms from surfaces. Chemical sanitizers, such as chlorine-based compounds, quaternary ammonium compounds (QACs), and peracetic acid, are commonly used to disinfect surfaces. Thermal treatments involve the application of hot water or steam to kill microorganisms [72].

Enzymatic cleaners have emerged as an innovative approach to biofilm control. These cleaners contain enzymes such as proteases, amylases, and DNases that specifically target and degrade components of the EPS matrix. By breaking down proteins, polysaccharides, and extracellular DNA, enzymatic cleaners facilitate the removal of biofilms from surfaces [73]. The application of dispersin B, an enzyme that hydrolyzes poly-N-acetylglucosamine, has shown significant biofilm removal on various surfaces [74].

Recent studies indicate that electrolyzed water, both alkaline and neutral, can offer a more sustainable and effective alternative to conventional cleaning-in-place (CIP) systems for biofilm removal on stainless steel surfaces. Unlike traditional CIP protocols, which often rely on aggressive chemicals and elevated temperatures, electrolyzed water exhibits antimicrobial activity due to its reactive oxygen species content while being less corrosive and environmentally damaging. Alkaline electrolyzed water facilitates the breakdown of organic residues and biofilm matrices, while neutral electrolyzed water contributes to microbial inactivation. These properties make electrolyzed water a promising candidate for routine sanitation protocols [75].

Bacteriophages, viruses that infect and lyse specific bacteria, offer a targeted approach to biofilm control. Phage therapy involves the application of bacteriophages to surfaces or products to reduce bacterial populations. Bacteriophages can penetrate biofilms and infect the bacteria within, leading to biofilm disruption and bacterial cell death. Recent advances have explored the use of bacteriophage-derived enzymes, such as endolysins, which can degrade bacterial cell walls and biofilm matrices. These enzymes have shown promise in controlling biofilms formed by pathogens like *Listeria monocytogenes* and *Salmonella enterica* [32,76].

Bacteriophage applications offer targeted biofilm control by infecting and lysing specific bacterial species within biofilms, reducing the risk of contamination and subsequent recalls. Surface modifications, such as the application of antimicrobial coatings, can prevent bacterial adhesion and biofilm formation, prolonging equipment lifespan and maintaining product quality [77]. Modifying the surfaces of food-processing equipment to prevent bacterial adhesion represents another innovative strategy for biofilm control. Such surface modifications include the application of antimicrobial coatings such as silver nanoparticles, copper alloys, or chitosan composites that exert bactericidal effects by disrupting cell membranes or generating reactive oxygen species. Additionally, engineering superhydrophobic surfaces can minimize microbial adhesion by creating water-repellent, low-surface-energy interfaces that inhibit bacterial contact and retention. These modifications primarily aim to reduce the initial stages of microbial attachment, thereby preventing biofilm establishment. Notably, the development of nanoscale surface textures and engineered surface roughness has been shown to significantly impair bacterial adhesion and biofilm formation on stainless steel and polymeric materials used in the food industry [3,18].

### 3.1. Disinfection Limitations in Mixed Biofilms

Conventional cleaning and sanitation methods face significant limitations in effectively removing biofilms from food-processing environments. The extracellular polymeric substances matrix of biofilms acts as a physical barrier, impeding the penetration of sanitizers and shielding embedded microorganisms from chemical and thermal treatments. Furthermore, mechanical cleaning may fail to reach critical zones in complex equipment, allowing biofilms to persist and recontaminate surfaces [78].

In meat and dairy-processing facilities, biofilms composed of *Listeria monocytogenes*, *Salmonella* spp., and *Pseudomonas* spp. have been found to be particularly resistant to standard cleaning procedures. These biofilms can persist on various surfaces, including stainless steel and plastic, even after routine sanitation efforts [79]. The persistence of such biofilms poses a major food safety risk, as they can serve as continuous sources of contamination and potential vectors for the horizontal gene transfer of resistance traits [80].

Recent studies have demonstrated that specific combinations of pathogens within multi-species biofilms can lead to synergistic resistance to cleaning agents and disinfectants. For example, co-cultures of *Listeria monocytogenes* with either *Pseudomonas* spp. or *Salmonella* spp. have been shown to produce biofilms with increased biomass, greater structural complexity, and significantly reduced susceptibility to enzymatic disruption. *Pseudomonas* spp., often acting as a pioneer colonizer, contributes heavily to EPS production, which acts as a physical shield for other pathogens, enhancing their survival under stress. Experimental studies have reported that such dual-species biofilms may exhibit two to three times higher resistance to standard sanitizers compared to their mono-species counterparts [81,82,83]. Similarly, *Staphylococcus aureus* and *Bacillus cereus* co-biofilms display increased sporulation and strong surface adhesion, making them particularly difficult to eliminate from stainless steel surfaces in dairy environments [84].

In a study involving 92 bacterial strains from dairy, meat, and egg-processing industries, researchers found that certain combinations of four species exhibited a biofilm mass more than 1.5 times greater than the sum of individual species’ biofilms. This synergy was particularly notable among dairy strains [41].

To address these challenges, emerging biocontrol strategies, including enzymatic formulations and bacteriophage (phage) cocktails, are being explored for their effectiveness against mixed-species biofilms. While mono-species *Listeria monocytogenes* biofilms respond well to single-enzyme cleaners or specific phage treatments, mixed-species biofilms show only partial disruption unless combined enzymatic and phage approaches are used.

Enzymatic cleaners targeting β-glucans are more effective against *Bacillus*-rich biofilms, while protease-based agents are more effective against *Staphylococcus*-dominated communities. Phage cocktails must be designed to target multiple species or combined with EPS-degrading enzymes and surfactants to penetrate protective matrices. Furthermore, surface engineering, such as the use of silver nanoparticle coatings or quorum sensing inhibitors, has shown selective efficacy, depending on whether the dominant organisms are Gram-positive or Gram-negative [85,86].

### 3.2. Regulatory and Compliance Considerations

Biofilm control in the food industry is governed by a combination of international standards and national regulations aimed at ensuring food safety and hygiene. One of the primary international standards is the ISO 22000 Food Safety Management System, which provides a framework for managing food safety risks, including those posed by biofilms. ISO 22000 integrates the principles of Hazard Analysis and Critical Control Points (HACCP) and emphasizes the importance of prerequisite programs (PRPs) to maintain hygienic conditions in food-processing environments [12].

The Codex Alimentarius, established by the Food and Agriculture Organization (FAO) and the World Health Organization (WHO), offers guidelines and codes of practice for food hygiene, including measures to control biofilm formation. These guidelines serve as a reference for national regulations and are recognized by the World Trade Organization (WTO) as international food safety standards [87].

In the United States, the Food Safety Modernization Act (FSMA) enacted by the Food and Drug Administration (FDA) mandates preventive controls for food facilities. Under the FSMA, facilities are required to implement risk-based preventive controls, which include sanitation procedures to address hazards such as biofilms. The act emphasizes proactive measures to prevent contamination rather than reactive responses [88].

Adherence to established hygiene standards is crucial for preventing biofilm formation and ensuring food safety. Implementing Good Manufacturing Practices (GMPs) and maintaining sanitary conditions in food-processing environments help minimize the risk of biofilm development. Regular cleaning and sanitation protocols, equipment maintenance, and employee training are essential components of these practices [89].

Failure to comply with hygiene standards can lead to persistent biofilm-related contamination, resulting in foodborne illnesses, product recalls, and economic losses. Therefore, food industry stakeholders must prioritize compliance with regulatory requirements and continuously monitor and improve their hygiene practices to mitigate the risks associated with biofilms [90].

## 4. Biofilms in Specific Food-Processing Environments

### 4.1. Dairy-Processing Facilities

Dairy-processing environments are particularly susceptible to biofilm formation due to the presence of nutrient-rich residues and favorable conditions for microbial growth. Moreover, proteinaceous deposits from milk (e.g., whey or casein layers) create conditioning films that facilitate microbial attachment and matrix formation. Frequently isolated biofilm-forming genera include *Bacillus* (e.g., *B. cereus*, *B. licheniformis*), *Staphylococcus* (*notably S. aureus*), *Pseudomonas*, *Enterococcus*, *Lactococcus*, and *Acinetobacter.* [91,92,93,94,95].

*Pseudomonas* spp. isolated from milk and dairy products frequently demonstrate a strong ability to form biofilms, which enhances their persistence in dairy environments and contributes to product spoilage. Studies show that 65–90% of *Pseudomonas* isolates from milk and dairy products are capable of biofilm formation, with some strains producing higher biofilm biomass at refrigeration temperatures (6–7 °C), which is relevant for dairy storage conditions. High biofilm producers are commonly found among *P. fluorescens*, *P. gessardii*, *P. koreensis*, and related subgroups. The presence of the adnA gene, associated with biofilm formation, was detected in a significant proportion of biofilm-producing strains though not universally across all species [94,96]. A mixed-species biofilm including *Pseudomonas fragi* and *Brochothrix* established on stainless steel allowed *L. monocytogenes* to integrate within 2 h and persist at ~2.4% of the community after 7 days, demonstrating its opportunistic and persistent behavior [95].

*Pseudomonas* spp. from dairy sources exhibit notable resistance to multiple antibiotics. Resistance rates are particularly high for aztreonam (60–73%), imipenem (28–95%), meropenem (13–37%), and ceftazidime (19%). Some studies report that 30–46% of isolates are multidrug-resistant (MDR) with multiple antibiotic resistance index (MARI) values exceeding 0.2 in a substantial fraction of strains. Whole-genome sequencing has revealed the presence of various resistance genes, including those encoding β-lactamases and efflux pumps, which contribute to intrinsic and acquired resistance [94,97,98,99].

Concurrently, spore-forming *Bacillus* spp. thrive in dairy facilities: isolates of *B. cereus*, *B. subtilis*, *B. licheniformis*, and *B. paralicheniformis* produce biofilms containing heat-resistant spores that withstand standard automated cleaning systems. These spores can survive pasteurization processes and lead to recurring contamination issues [100]. Notably, in the presence of skim milk, thermophilic *Bacillus* spores adhere to stainless steel surfaces 10–100× more efficiently; biofilm-derived spores display up to 2.3 log higher resistance to mechanical and chemical cleaning agents compared to non-dairy strains [101].

*S. aureus* forms biofilms on stainless steel, various rubbers (Buna-N, EPDM, silicone), glass, polycarbonate, and PVC materials commonly found in milking equipment. Biofilm formation is especially strong on Buna-N rubber, but it occurs on most surfaces tested with some strain and material-specific differences [102,103]. Many *S. aureus* isolates from dairy environments carry genes (icaADBC, bap, fnbA/B, clfA/B, sar, sigB) that promote biofilm formation and adhesion, contributing to their persistence and virulence [103].

### 4.2. Meat-Processing Facilities

Meat-processing environments offer ideal conditions for biofilm formation due to constant exposure to organic matter, moisture, and temperature variations. These factors favor bacterial adhesion and growth on surfaces such as stainless steel, rubber, and plastic. Biofilms in meat facilities often harbor mixed-species communities, prominently including *Listeria monocytogenes*, *Salmonella* spp., *Escherichia coli*, and *Pseudomonas* spp., which are frequently implicated in cross-contamination and persistent hygiene failures [104,105,106].

The role of *Pseudomonas* spp. in biofilm resilience is particularly noteworthy. Psychrotrophic *Pseudomonas* species dominant in meat environments enhance the survival and disinfectant tolerance of cohabiting pathogens within mixed-species biofilms [105].

*Pseudomonas* species, especially *P. fragi* and *P. aeruginosa*, can create biofilms that support the survival and colonization of *Campylobacter*, including *C. jejuni*, in food-processing environments. The presence of *Pseudomonas* biofilms enhances *Campylobacter’s* ability to persist under otherwise hostile conditions, such as high oxygen levels and low temperatures. *Pseudomonas* spp. help *Campylobacter* survive in atmospheric oxygen, which is a condition that is normally lethal to this microaerophilic pathogen. This is likely due to metabolic interactions and the creation of microenvironments within the biofilm that reduce oxygen stress [106,107,108].

Environmental studies further reveal that microbial communities in drains and floor surfaces contribute significantly to biofilm resilience. Long-term resident microbiota, such as *Pseudomonas* spp. and *Acinetobacter*, stabilize environmental niches, enhancing *Listeria* persistence [109]. Wang et al. (2024) reported that sanitation cycles in beef plants may paradoxically select for denser, more resistant biofilms in drainage systems, especially protecting *Salmonella* against quaternary ammonium compounds [110]. Environmental isolates from meat plant drains can modulate *E. coli* O157:H7 biofilm formation, altering its resistance profile depending on the species composition [111].

*Listeria monocytogenes* remain a key concern due to their ability to persist in meat-processing environments despite routine sanitation. Strains isolated from meat plants have shown strong biofilm-forming capacities linked to genetic markers associated with persistence and stress resistance. Floor drains, conveyor systems, and cutting surfaces are particularly implicated as reservoirs for *Listeria* with microbial communities stabilizing in these niches over time [112].

The architecture of meat-processing facilities with joints, seals, porous materials, and drainage systems creates ideal harborage sites for biofilms. Water hoses and drains are particularly problematic, as biofilms here can detach and contaminate cleaned surfaces or products [113].

Similarly to the dairy industry, bacteria embedded in biofilms within meat-processing environments can develop and maintain antimicrobial resistance, which is a phenomenon amplified by environmental stressors and repeated exposure to sublethal concentrations of disinfectants [110].

A study by Rossi et al. (2019) investigated *C. jejuni* isolates from poultry slaughterhouses and found that biofilm-growing cells exhibited significantly higher resistance to key antibiotics, ciprofloxacin, erythromycin, colistin, and meropenem compared to their free-floating counterparts. Planktonic cells showed varied resistance, but once incorporated into biofilms (often in the presence of chicken juice), all tested strains required antibiotic concentrations ≥ 32 mg/L to inhibit, underlining the protective role of biofilms in clinical and food-processing contexts [114].

Manafi et al. (2020) investigated the antibiotic resistance profiles and biofilm-forming abilities of *Salmonella* serotypes isolated from beef, mutton, and meat contact surfaces in retail settings. The study revealed that a significant proportion of isolates exhibited multidrug resistance (MDR), particularly to tetracycline, streptomycin, and ampicillin. Furthermore, over 70% of the *Salmonella* isolates demonstrated moderate to strong biofilm formation on polystyrene surfaces [115].

Romeo et al. (2025) reported that *Listeria monocytogenes* isolates from meat-processing plants often carried resistance to tetracycline and erythromycin among diverse genetic lineages. The study highlighted that resistant strains were often associated with strong biofilm-forming ability [116].

Emerging control strategies in meat facilities focus on the application of tailored biocontrol methods, such as enzymatic cleaners, bacteriophage cocktails, and advanced surface coatings. These interventions aim to disrupt multi-species biofilms and prevent cross-contamination, supplementing traditional cleaning-in-place systems that may be insufficient on their own.

### 4.3. Seafood-Processing Facilities

Seafood-processing environments offer highly favorable conditions for biofilm formation, which is driven by persistent humidity, high salinity, and nutrient-rich residues that promote bacterial adhesion and matrix production on processing surfaces. Unlike meat and dairy sectors, where protein and fat residues dominate, seafood environments uniquely select for halotolerant species like *Vibrio* spp. and *Listeria monocytogenes*, which are both notorious for their persistence and resistance within processing lines [117]. The adaptation of *Vibrio parahaemolyticus* to sublethal disinfectant exposure, especially benzalkonium chloride (BAC), has been shown to induce biofilm formation and the emergence of viable but non-culturable cells, complicating their detection and eradication in seafood facilities [118]. Similarly, *L. monocytogenes* biofilms grown under high-humidity, saline conditions display enhanced resistance to sanitizers, which is a phenomenon that has been partly mitigated through the application of fucoidan, an algal-derived compound with quorum-sensing inhibitory effects [42].

To address these challenges, combined disinfection strategies have been explored, such as sodium hypochlorite coupled with UV-C irradiation, which has demonstrated improved efficacy against *V. parahaemolyticus* biofilms on seafood-processing lines [119]. Furthermore, peracetic acid (PAA) has proven effective in controlling planktonic *V. parahaemolyticus* populations at concentrations as low as 0.005%, suggesting potential for integration into routine sanitation protocols in seafood environments [120]. Biofilm structure and resilience are also influenced by atmospheric conditions; studies show that *V. parahaemolyticus* and *L. monocytogenes* biofilms alter their architecture and EPS composition in response to varying oxygen levels, highlighting the importance of environmental management during processing and storage [121].

Although data on antimicrobial resistance (AMR) in seafood-associated biofilms remain limited compared to the meat and dairy sectors, studies suggest that biofilm-embedded *Vibrio* strains have been found to harbor resistance genes against β-lactam antibiotics (e.g., blaTEM, blaCTX-M-55, blaAmpC), aminoglycosides (aac(6’)-Ib, aac(3)-IV), tetracycline (tetA), and fluoroquinolones (qnrA) [121,122,123,124,125].

### 4.4. Beverage Industry

The beverage sector, including breweries, soft drink manufacturing, and bottling facilities, faces persistent biofilm challenges associated with the continuous processing of sugar-rich liquids in moist environments. Spoilage microorganisms such as *Lactobacillus* spp., *Pediococcus* spp., and *Acetobacter* spp. readily form biofilms on pipelines, storage tanks, and filling equipment, leading to product spoilage, off-flavors and recurring contamination issues [126].

Complex biofilms in commercial draft beer-dispensing systems including *Acetobacter* and *Lactobacillus* species persist even after routine cleaning, posing a hazard to product quality and consumer safety [127,128].

Beverage-processing lines are particularly susceptible to biofilm formation due to the widespread use of narrow-diameter piping with complex loops, junctions, and dead-end sections found in carbonation circuits, syrup-dosing systems, and beverage dispensers. These architectural features create stagnant zones beyond the reach of cleaning solutions, fostering conditions for biofilm establishment and persistence. Bottling and filling machines also present critical control points, as valves, nozzles, and gaskets serve as frequent sites of residue buildup and microbial adhesion [126]. Moreover, the moderate operating temperatures (20–30 °C) and the acidic or sugar-rich nature of many beverages further support the growth of spoilage organisms. Inadequate flow rates and insufficient turbulence during CIP cycles reduce cleaning efficacy, particularly in narrow or irregular piping, making biofilm removal challenging [129].

Beyond microbiological risks, biofilms in beverage processing can cause serious technical issues, including the microbially influenced corrosion (MIC) of stainless-steel equipment and pipelines. The biofilm matrix promotes localized corrosion by creating anaerobic microenvironments and facilitating the accumulation of corrosive metabolic by-products, such as organic acids and sulfides. This results in equipment degradation, reduced heat transfer efficiency, and increased maintenance costs [130].

### 4.5. Bakery and Ready-to-Eat Foods

Bakery and ready-to-eat (RTE) food production environments face contamination risks linked to the high carbohydrate content, low water activity zones, and post-processing handling stages typical of these industries. These environments promote the persistence of specific pathogens such as *Staphylococcus aureus* and *Listeria monocytogenes*, particularly during cooling, slicing, and packaging operations [131].

A critical risk factor in bakery lines is post-baking contamination. Although baking itself provides an effective kill step, subsequent exposure to ambient air, conveyor belts, and slicing equipment can reintroduce pathogens. Studies show that L. monocytogenes is present in a small but significant proportion of RTE foods. For example, a large-scale survey in Poland found contamination in 0.1% of RTE products with strains often linked to human listeriosis and showing resistance to multiple antibiotics [132]. According to a 2021 RASFF notification, *Listeria monocytogenes* contamination was detected in deep-frozen bakery products originating from Ukraine and distributed across several EU countries. The notification, issued by Hungary, resulted in the withdrawal of the affected products from the market, highlighting the importance of strict post-processing control measures in the bakery sector [133].

Additionally, bakery products incorporating creams, custards, or fillings face dual risks of contamination with psychrotrophic pathogens and spoilage organisms. Epidemiological data have identified Listeria monocytogenes contamination in refrigerated cream-filled bakery products. For instance, a Croatian survey detected the pathogen in 6.25% of cake samples—many containing cream or custard fillings [134].

Many countries set a limit of 100 CFU/g for L. monocytogenes in low-risk RTE foods, while others (like the US) enforce zero tolerance, leading to recalls for any detection [135].

Humidity and condensation within bakery cooling tunnels and packaging zones further exacerbate contamination risks. Controlled studies demonstrated that condensation droplets on cooled surfaces can facilitate bacterial transfer rates up to 4-log higher compared to dry surfaces, particularly favoring *L. monocytogenes* adhesion on stainless steel and polyethylene materials [136].

A case study by Ribeiro and Clérigo (2017) assessed *Staphylococcus aureus* colonization among bakery workers and revealed that 40% of employees were carriers with 25% of isolates identified as methicillin resistant (MRSA). The study also highlighted lapses in hygiene practices, such as the inconsistent use of personal protective equipment, emphasizing the role of food handlers as a potential source of *S. aureus* contamination in bakery environments [137]. Hait et al. (2012) reported a *Staphylococcus aureus* outbreak linked to a bakery in Illinois, where contaminated cream-filled pastries caused foodborne illness in over 100 individuals. The investigation revealed that inadequate refrigeration and poor hygiene practices allowed *S. aureus* to proliferate and produce enterotoxins in the bakery products [138]. Necidová et al. (2022) evaluated the growth of *S. aureus* and enterotoxin production in delicatessen and fine bakery products under various storage conditions. The study showed that *S. aureus* could multiply rapidly on bakery products stored at ambient temperatures, reaching levels sufficient for enterotoxin production within a short time [139].

Addressing these sector-specific challenges requires integrating targeted measures such as the real-time monitoring of slicing equipment, humidity control in packaging zones, and rigorous hygiene protocols for post-processing stages. Antimicrobial coatings and treated surfaces, including belts and blades, have demonstrated the ability to reduce bacterial contamination and cross-contamination in food-processing environments. Reported reductions in microbial load can be substantial, but the exact percentage varies depending on the antimicrobial agent and application method [136,140]. Contact-active antimicrobial surfaces (such as cationic polymers) are effective in the short term but may lose activity as organic material or bacterial debris accumulates, shielding the active surface from further contamination [141,142,143]. While pilot studies and laboratory tests show promising reductions in contamination (sometimes up to 80%), large-scale, long-term implementation faces challenges such as cost, durability, and maintaining antimicrobial activity over time [140,141,142].

### 4.6. Fresh Produce Handling

Fresh produce handling presents unique challenges regarding biofilm formation, especially because fruits and vegetables are often consumed raw or minimally processed. Pathogens such as *Escherichia coli* O157:H7, *Salmonella enterica*, and *Listeria monocytogenes* are frequently implicated in produce-related outbreaks. The irregular topography of produce such as stomata, lenticels, trichomes, and cuticular cracks creates protective niches that facilitate initial bacterial adhesion and subsequent biofilm development [144,145,146].

During harvesting, contaminated irrigation water, soil, or handling equipment can introduce biofilm-forming bacteria onto produce surfaces. A systematic review of 277 fresh produce associated outbreaks in Europe and North America found that vegetables accounted for 34.1% of outbreaks in Europe and 47.4% in North America with contaminated irrigation water frequently identified as a contributing factor [147]. Supporting this, a 2013 outbreak investigation in Sweden linked *E. coli* O157:H7 contamination of lettuce directly to irrigation water used on the farm [148].

Patel et al. (2013) investigated *Salmonella* survival on spinach irrigated with contaminated water. While *Salmonella* levels fell below detection at 24 h when water contamination was low (<126 CFU/100 mL), strains with strong biofilm-forming ability persisted significantly longer, up to 35 days, when contamination was high [149].

Kim et al. (2023) investigated the internalization of *Salmonella enterica* into leafy vegetables under post-harvest conditions, focusing on factors such as temperature, humidity, and wash water contamination. The study demonstrated that *Salmonella* can rapidly adhere to leaf surfaces and internalize through natural openings like stomata and cut edges. Higher storage temperatures and contaminated wash water significantly increased the risk of internalization and persistence [42].

Sanitization procedures such as chlorine washes and peracetic acid treatments often fail to eradicate biofilms on produce. Chhetri et al. (2019) demonstrated that *E. coli* O157:H7 and *Listeria monocytogenes* biofilms on spinach leaves exposed to 100 ppm free chlorine exhibited a markedly reduced log reduction compared to free-floating cells [150].

Similarly, *Salmonella* biofilms have been shown to persist on tomato surfaces despite standard sanitization procedures. Studies highlighted the capacity of *Salmonella* to embed within natural surface crevices, such as stem scars, allowing the pathogen to evade surface disinfection methods commonly applied post-harvest [151].

The efficacy of sanitizers against foodborne pathogens is influenced by the type of surface and environmental conditions. Park and Kang (2018) demonstrated that chlorine dioxide (ClO_2_) gas was significantly more effective at lower temperatures. When applied at 15 °C, ClO_2_ achieved up to a 3-log reduction in *Escherichia coli* O157:H7, *Salmonella Typhimurium*, and *Listeria monocytogenes* on spinach leaves, tomatoes, and stainless steel surfaces. In contrast, treatment at 25 °C resulted in markedly lower pathogen reductions. The authors also observed that glass surfaces allowed for higher pathogen inactivation under similar conditions [152].

Storage conditions further influence biofilm formation on produce. Modified atmosphere packaging (MAP), commonly used to prolong shelf life, can inadvertently support the persistence of *Listeria monocytogenes* biofilms by creating microaerophilic environments favorable for their growth on fresh-cut vegetables even at temperatures as low as 4 °C [153].

Figure 1 illustrates the distribution of biofilm-associated contamination across major food sectors, illustrating the need for sector-specific strategies to manage and mitigate biofilm formation and its consequences on food safety [91,92,93,94,95,96,97,98,99,100,101,102,103,104,105,106,107,108,109,110,111,112,113,114,115,116,117,118,119,120,121,122,123,124,125,126,127,128,129,130,131,132,133,134,135,136,137,138,139,140,141,142,143,144,145,146,147,148,149,150,151,152,153,154,155,156,157,158].

## 5. Economic Impact Analysis

### 5.1. Impact on Food Safety and Quality

Biofilms in food-processing environments lead to significant economic burdens due to their role in contamination, equipment damage, and increased operational costs. Persistent biofilms can harbor pathogenic microorganisms, leading to foodborne illnesses and necessitating costly product recalls. For instance, biofilm-associated contamination has been implicated in numerous foodborne illness outbreaks, resulting in substantial financial losses for companies, including legal fees, lost sales, and reputational damage [159].

Biofilms contribute to equipment degradation through corrosion and fouling, leading to increased maintenance costs and potential equipment replacement. The presence of biofilms on processing equipment surfaces can impair heat transfer efficiency and increase energy consumption, further escalating operational expenses [160].

Cleaning and sanitation efforts to control biofilms also incur additional costs. The resilience of biofilms to standard cleaning agents necessitates more frequent and intensive cleaning protocols, increasing labor, water, and chemical usage. These enhanced cleaning requirements can disrupt production schedules and reduce overall efficiency [7].

Implementing advanced biofilm control strategies, such as enzymatic cleaners, bacteriophage applications, and surface modifications, can entail higher initial investments. However, these measures have been shown to effectively reduce biofilm formation and persistence, leading to long-term cost savings. For example, the use of enzymatic cleaners can enhance biofilm removal efficiency, decreasing the frequency and intensity of cleaning cycles [161].

While the upfront costs of these advanced interventions may be higher, the potential savings from reduced product losses, lower maintenance expenses, and minimized downtime can offset the initial investments. A comprehensive cost–benefit analysis should consider these long-term economic advantages when evaluating biofilm control strategies.

Quantitative cost analyses suggest that the implementation of enzymatic and bacteriophage-based treatments in food-processing environments is promising but requires careful financial planning. Enzymatic cleaners, particularly those targeting polysaccharides or proteins in EPS matrices, have an estimated cost ranging from 3 to 5 Euros per liter, depending on formulation and supplier. Based on pilot-scale studies in dairy-processing facilities, a single enzymatic treatment cycle, requiring 10–15 L per 1000 L of CIP solution, incurs operational costs between 30 and 75 euros per cleaning session, which is slightly higher than traditional caustic cleaning but with measurable reductions in water consumption and residue levels [162]. Compared to enzymatic cleaners, commercially available phage products are priced at approximately 0.09–0.27 Euros per pound (0.45 kg) of treated food particularly in meat-processing environments [163]. Despite the higher upfront cost, phage treatments have been shown to achieve reductions of up to 4 log units in microbial populations, thereby extending the intervals between sanitation cycles and significantly reducing overall contamination levels [164]. However, both methods face economic challenges in scalability, especially for high-throughput facilities, where repeat applications and storage stability (particularly cold-chain requirements for phages) add recurring operational costs.

As shown in Figure 2 and Table 4, biofilm-related economic losses vary substantially across continents with Europe and North America experiencing the highest total costs due to large-scale, regulated food industries and frequent recalls. In contrast, Asia and Africa face unique challenges tied to infrastructure and environmental conditions [159,165,166,167,168,169,170,171,172,173,174,175,176,177,178,179,180,181,182,183,184,185,186,187,188,189,190,191,192,193,194,195].

### 5.2. Consumer Health Implications

The public health risks posed by biofilms in the food industry are multifaceted, extending beyond contamination to more serious implications such as persistent foodborne outbreaks and antimicrobial resistance (AMR). While previous sections have outlined the resilience of biofilms to standard sanitization and their role in harboring pathogens, it is critical to emphasize the downstream impact on human health and the broader healthcare system [1]. Biofilm-associated pathogens such as *Listeria monocytogenes*, *Salmonella* spp., *Escherichia coli* O157:H7, and *Cronobacter sakazakii* have been linked to severe illnesses with high morbidity and mortality particularly in vulnerable populations including infants, pregnant women, the elderly, and immunocompromised individuals [61,196,197,198]. These microorganisms, once shielded within biofilms, exhibit enhanced resistance to antibiotics and disinfectants, leading to prolonged survival on food contact surfaces and in processing environments [199].

The resilience of biofilms to cleaning agents means that even with standard sanitation protocols, these pathogens can remain on surfaces and contaminate subsequent batches of food. This persistent contamination underscores the importance of effective biofilm control measures to protect public health [1].

Educating consumers about proper food-handling practices is crucial in preventing foodborne illnesses associated with biofilm-forming pathogens. Public awareness campaigns can inform individuals about the risks of consuming contaminated foods and the steps they can take to minimize these risks. For example, the Partnership for Food Safety Education’s ‘Fight BAC!’ campaign, operating in the United States, emphasizes four key practices: clean, separate, cook, and chill. These guidelines help consumers understand the importance of hygiene, preventing cross-contamination, cooking foods to safe temperatures, and proper refrigeration [200].

Moreover, public education initiatives can highlight the significance of purchasing food from reputable sources, checking expiration dates, and being aware of food recalls. By empowering consumers with knowledge, these programs play a vital role in reducing the incidence of foodborne illnesses linked to biofilm-associated pathogens [201].

## 6. Conclusions

Biofilms present a persistent and complex challenge across all sectors of the food industry. These structured microbial communities not only facilitate the survival of pathogenic and spoilage organisms under harsh conditions but also significantly reduce the efficacy of conventional cleaning and disinfection protocols. Their ability to form on a wide range of surfaces including stainless steel, plastics, and rubber enables the long-term contamination of food-processing environments, leading to serious public health implications and considerable economic losses.

As highlighted in this review, biofilm-associated risks are particularly critical in dairy, meat, seafood, beverage, bakery, and fresh produce industries, where environmental factors such as moisture, temperature, and nutrient residues favor rapid microbial adhesion and matrix formation. Moreover, the role of multi-species biofilms, quorum sensing, and horizontal gene transfer in enhancing biofilm resilience and antimicrobial resistance underscores the need for multidisciplinary strategies.

Current detection and control measures, while useful, are often insufficient for complete biofilm eradication. Therefore, future directions must prioritize the development of real-time monitoring systems, natural and enzymatic biofilm disruptors, innovative surface modifications, and integration of biofilm control within existing food safety management systems such as HACCP and ISO 22000.

Furthermore, ongoing education, regulatory reinforcement, and investment in biofilm research are essential for minimizing risks and improving food safety. A proactive, evidence-based approach tailored to specific processing environments and microbial threats will be critical in reducing the prevalence and impact of biofilms in the global food supply chain.

## Figures and Tables

**Figure 1 microorganisms-13-01805-f001:**
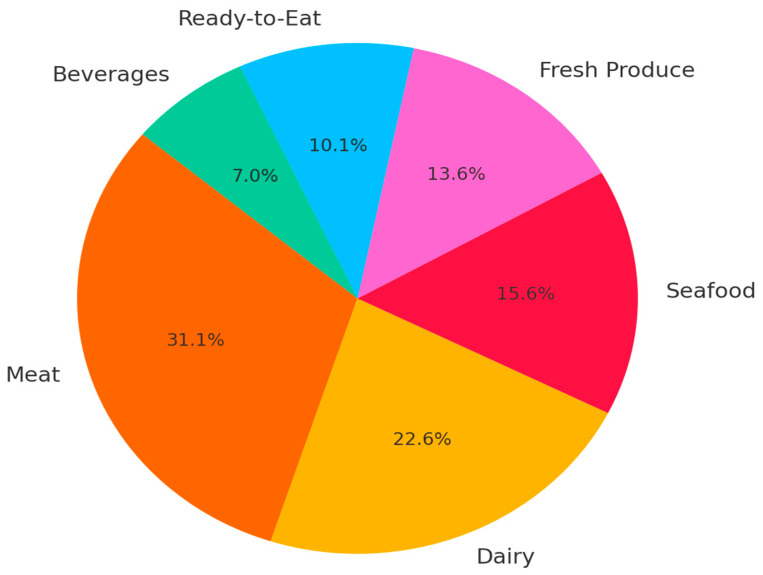
Distribution of biofilm-associated contamination across food sectors. Legend: Sectoral risk distribution based on literature evidence of biofilm occurrence and contamination trends (references [91,92,93,94,95,96,97,98,99,100,101,102,103,104,105,106,107,108,109,110,111,112,113,114,115,116,117,118,119,120,121,122,123,124,125,126,127,128,129,130,131,132,133,134,135,136,137,138,139,140,141,142,143,144,145,146,147,148,149,150,151,152,153,154,155,156,157,158]). Exact percentages are conceptual and based on combined qualitative data from the reviewed studies.

**Figure 2 microorganisms-13-01805-f002:**
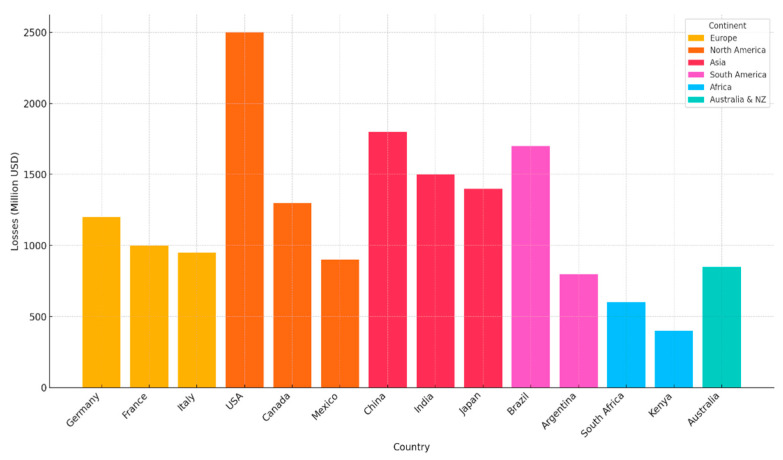
Estimated economic losses from biofilm contamination by country. Legend: Values are conceptual estimates derived from qualitative synthesis and sector studies, intended to illustrate cross-country comparison data from the reviewed studies [159,165,166,167,168,169,170,171,172,173,174,175,176,177,178,179,180,181,182,183,184,185,186,187,188,189,190,191,192,193,194,195].

**Table 1 microorganisms-13-01805-t001:** Biofilm-associated pathogens.

Pathogen	Health Implication	Severity	At-Risk Group	Biofilm Relevance	Reference
*Pseudomonas* *aeruginosa*	Opportunistic infections	Severe in vulnerable	Immunocompromised	Highly resistant to disinfectants	[1]
*Bacillus cereus*	Vomiting, diarrhea	Mild to moderate	General population	Spore-former; survives cooking	[1]
*Vibrio* spp.	Cholera, gastroenteritis	Severe	Coastal communities, travelers	Survives in marine biofilms	[43,44]
*Cronobacter* *sakazakii*	Sepsis, meningitis	Severe	Infants (neonates)	Survives in powdered infant formula	[45]
*Yersinia* *enterocolitica*	Enteritis	Moderate	Children	Cold-tolerant biofilms	[46]
*Listeria* *monocytogenes*	Listeriosis	Severe	General population	Biofilm protects bacteria; persists on surfaces	[1]
*Salmonella* spp.	Salmonellosis	Moderate to severe	General population	Forms resistant biofilms on various surfaces	[1]
*Escherichia coli* O157:H7	Hemorrhagic colitis	Severe	General population	Biofilms enhance survival on meat and produce	[1]
*Staphylococcus aureus*	Food poisoning	Mild to moderate	General population	Biofilms increase cleaning resistance	[1]
*Campylobacter* spp.	Gastroenteritis	Moderate	General population	Biofilms promote persistence in food/water	[1]

**Table 2 microorganisms-13-01805-t002:** Foodborne outbreaks linked to biofilm-forming bacteria across Europe.

Country	Year	Product	Bacteria	Setting	People Affected	Control Measure	Reference
France	2013	Bottled mineral water	*Salmonella enterica*	Bottling plant	Multiple cases	Plant sanitation	[47,48]
Germany	2018	Blood sausage	*Listeria monocytogenes*	Processing plant	47 cases	Recall	[49]
Spain	2019	Chilled pork products	*Listeria monocytogenes*	Meat-processing facility	>200 cases, 3 deaths	Recall	[50]
Italy	2020	Hospital-prepared meals	*Listeria monocytogenes*	Hospital kitchen	4 patients	Equipment sanitation	[51]
Greece	2021	School meals	*Clostridium perfringens* *Bacillus cereus*	Catering service	30 students	Improved hygiene	[52]
Poland	2022	Various food samples	*Listeria monocytogenes* *Salmonella* spp. *Escherichia coli*, *Bacillus cereus*	Retail food samples	Not specified	Surveillance	[53]

**Table 4 microorganisms-13-01805-t004:** Regional impact of biofilm-associated foodborne pathogens in the global food industry.

Continent	Key Food Sectors Affected	Estimated Annual Losses	Most Common Biofilm-Forming Pathogens	Typical Control Challenges	Notable Outbreaks/Case Studies
Europe	Dairy, Meat, Seafood, Fresh Produce	EUR 5–6 billion	*Listeria monocytogenes*, *Salmonella* spp.	Multi-species resistance, outdated equipment	2018 frozen vegetable outbreak (EU-wide, Hungary origin) [188]
North America	Meat, Ready-to-Eat, Dairy	USD 7–8 billion	*Listeria* spp., *E. coli* O157:H7, *S. aureus*	Compliance variation across processors	Blue Bell Creameries recall (USA, 2015) [165,189]
South America	Poultry, Produce, Seafood	USD 1.5–2 billion	*Salmonella* spp., *Pseudomonas* spp.	Tropical climate favors rapid growth, sanitation variability	Poultry-linked *Salmonella* spp. outbreaks (Brazil) [175]
Asia	Seafood, Fresh Produce, Street Foods	USD 3–4 billion	*Vibrio* spp., *Salmonella* spp., *Shigella* spp.	Water sanitation, informal processing sectors	Street food outbreaks (India, Southeast Asia) [190]
Africa	Dairy, Poultry, RTE foods	>USD 1 billion	*E. coli*, *Listeria* spp., *Campylobacter*	Resource-limited monitoring, limited traceability	Milk-related outbreaks in East Africa [192,193]
Australia New Zealand	Meat, Dairy, Fresh Produce	~USD 800 million	*Listeria monocytogenes*, *Salmonella enterica*	Long distribution chains, refrigeration gaps in remote areas	*Listeria* spp. in cantaloupe (Australia, 2018) [195]

Legend: RTE—ready to eat. Values are conceptual estimates derived from qualitative synthesis and sector studies intended to illustrate cross-country comparison data from the reviewed studies [159,165,166,167,168,169,170,171,172,173,174,175,176,177,178,179,180,181,182,183,184,185,186,187,188,189,190,191,192,193,194,195].

## Data Availability

No new data were created or analyzed in this study.

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
