# Peer review of "Persistent Threats: A Comprehensive Review of Biofilm Formation, Control, and Economic Implications in Food Processing Environments"

_microorganisms, 2025, doi:10.3390/microorganisms13081805_

Round 1
Reviewer 1 Report
Comments and Suggestions for Authors
This article examines the mechanisms of biofilm formation, control strategies, and their economic impact in food processing environments. The article analyzes the mechanism of biofilm formation and its resistance to detergents and sanitizers. Also, biofilm detection techniques such as PCR, fluorescence microscopy and biosensors are investigated and control strategies such as enzymatic cleaners, phage therapy and surface modification are discussed. In addition, the article emphasizes the potential threat of biofilms to food safety and public health, as well as the economic burden they pose on a global scale. However, the following questions remain:
- It is recommended to standardize and beautify the formatting of charts and tables in the text.
- Table 3 and Table 4 are duplicated in the article, please double check.
- Although the existence of multi-species biofilms is mentioned in the article, there is less discussion on the specific ecological relationships between these microorganisms (e.g., competition, symbiosis, etc.) and their effects on biofilm structure and function.
- Although synergistic effects are mentioned, in-depth analysis of differences in resistance and specific control options for different species combinations is lacking.
- Challenges (e.g., cost, stability, etc.) for industrial application of emerging technologies are discussed superficially and lack specific data support.
- Although the advantages and disadvantages of detection techniques such as PCR are presented, comparative data on the effectiveness of different techniques applied in real food processing scenarios are lacking.
Author Response
Reviewer 1
We sincerely thank the reviewer for the constructive comments and insightful suggestions, which have greatly contributed to improving the clarity, accuracy, and overall quality of our manuscript. We have addressed all points carefully and revised the text accordingly.
- It is recommended to standardize and beautify the formatting of charts and tables in the text.
Thank you for the suggestion. All tables and figures have been revised.
- Table 3 and Table 4 are duplicated in the article, please double check.
We appreciate your careful observation; there was a confusion regarding were to place table 3 in section 2.6. Detection and Monitoring Techniques or 3. Control and Prevention Strategies. We have decided to place it in section 2.6.
- Although the existence of multi-species biofilms is mentioned in the article, there is less discussion on the specific ecological relationships between these microorganisms (e.g., competition, symbiosis, etc.) and their effects on biofilm structure and function.
Thank you for highlighting this gap. The section on Multi-species biofilms has been revised to include a discussion on ecological interactions such as commensalism, mutualism, syntrophy, and competitive exclusion.
- Although synergistic effects are mentioned, in-depth analysis of differences in resistance and specific control options for different species combinations is lacking
Thank you for the suggestion. An additional section „3.1 Disinfection Limitations in Mixed Biofilms” has been inserted to cover the missing information
- Challenges (e.g., cost, stability, etc.) for industrial application of emerging technologies are discussed superficially and lack specific data support.
This point has been addressed in the Economic Impact Analysis section.
- Although the advantages and disadvantages of detection techniques such as PCR are presented, comparative data on the effectiveness of different techniques applied in real food processing scenarios are lacking
We have addressed this by including several current studies on the topic (L346-364 )

Reviewer 2 Report
Comments and Suggestions for Authors
This article provides a comprehensive review of the literature on the conditions for the formation and importance of biofilms in various areas of the food industry. In my opinion, the study could benefit from expanding some chapters or subchapters. This can be done because the 14 pages of the review article give you the opportunity to supplement the content.
Below are my comments and questions regarding the text:
1. In many places in the text, there are sentences whose context is identical to the reader's. The first one is lines 48-51, which are very similar to the text on lines 41-44.
2. Chapter 2 would benefit from a graphical diagram of biofilm formation that would make this fragment more attractive.
3. Line 84: The abbreviation EPS appears here, and it should be explained here. It is only expanded on on line 153.
4. Line 118: Please indicate where these specific receptors are located.
5. Lines 136-138: Are there any known mechanisms for "disrupting QS signaling"? If so, please indicate them here.
6. Lines 145-152: The description of all these mechanisms is too general. The reader needs more detail. 7. Lines 162-164: I believe it's unnecessary to summarize at the end of each chapter/subsection. There are conclusions, after all, that are also a summary, but of the whole. This way, repetition can be avoided.
8. I believe that subsection 2.3 should not be included in chapter 2; it is also unnecessary.
9. Lines 166-172: another repetition of the context.
10. Line 171: Latin names of microorganisms are italicized.
11. Lines 177-181: another repetition of previously described content.
12. Lines 182-186: repeating content from lines 51-53. 13. Lines 199-203: I'm not sure what the connection between biofilm mass and the phenomenon of synergy is? Please explain this statement in more detail or write it in a way that's more understandable to the reader.
14. Lines 215-218: Again, a repetition of the context described earlier.
15. I believe that Chapter 2.5 requires additional content.
16. Table 2: I believe the column titled "Control measure" should be removed, as these issues appear later in the article.
17. Tables 2, 3, 4, 5, and Figure 2: They lack any references to the literature from which they were based. Please provide additional information.
18. I expected more details after Chapter 4, which shouldn't be difficult to provide. Literature on this topic already exists, so please expand on this chapter. This will be crucial for food technologists, and providing a single example for each industry is insufficient.
19. Lines 315-327: Repeat about bacteriophages.
20. Lines 388-395: Please clarify whether this Listeria belongs to this group of raw materials/processed products, or if this is a more general comment. 21. Lines 395-399: This is the third time I've mentioned planktonic forms, which are less resistant than biofilms :-(
22. Line 405: Italics are required for Latin names of microorganisms.
23. Line 487: Please provide information in which country the "Partnership for Food Safety Education's 'Fight BAC!'" operates.
24. Line 493: What is the significance of "being aware of food recalls" for consumers? I can't connect this with biofilms and their importance to human health, because that's what was discussed in this subsection.
25. Cited literature: Please ensure that all cited literature is recorded in a uniform manner. They should comply with the journal's requirements. Currently, there is a certain amount of discretion.
Author Response
Reviewer 2
We sincerely thank the reviewer for the constructive comments and insightful suggestions, which have greatly contributed to improving the clarity, accuracy, and overall quality of our manuscript. We have addressed all points carefully and revised the text accordingly.
- In many places in the text, there are sentences whose context is identical to the reader's. The first one is lines 48-51, which are very similar to the text on lines 41-44.
We thank the reviewer for highlighting the repetitions. We have carefully revised the manuscript and removed sentences whose context is identical to ensure that each section adds value to the reader.
- Chapter 2 would benefit from a graphical diagram of biofilm formation that would make this fragment more attractive.
We appreciate the reviewer’s suggestion to include a graphical diagram of biofilm formation in Chapter 2. While we agree that such a visual would enhance the section, we are currently unable to provide a suitable figure. However, we have ensured that the text clearly describes each stage of biofilm development to support the reader’s understanding in the absence of a graphic.
- Line 84: The abbreviation EPS appears here, and it should be explained here. It is only expanded on line 153.
We appreciate your careful observation; we have made the corresponding change.
- Line 118: Please indicate where these specific receptors are located.
We have added the specific information.
- Lines 136-138: Are there any known mechanisms for "disrupting QS signaling"? If so, please indicate them here.
Information regarding the disruption of QS signaling was added according to the reviewer request
- Lines 145-152: The description of all these mechanisms is too general. The reader needs more detail.
A detailed description of the mechanisms was added
- Lines 162-164: I believe it's unnecessary to summarize at the end of each chapter/subsection. There are conclusions, after all, that are also a summary, but of the whole. This way, repetition can be avoided.
We agree with this comment and removed any repeating summaries.
- I believe that subsection 2.3 should not be included in chapter 2; it is also unnecessary.
We agree with this comment and have removed this section
- Lines 166-172: another repetition of the context.
We agree with this comment and have made the necessary changes
- Line 171: Latin names of microorganisms are italicized.
We agree with this comment and have made the necessary changes
- Lines 177-181: another repetition of previously described content.
We agree with this comment and have made the necessary changes
- Lines 182-186: repeating content from lines 51-53.
We agree with this comment and have made the necessary changes
- Lines 199-203: I'm not sure what the connection between biofilm mass and the phenomenon of synergy is? Please explain this statement in more detail or write it in a way that's more understandable to the reader.
The statement was detailed and the following paragraph was inserted: The phenomenon of synergy in biofilms refers to a situation where the combined biofilm production of multiple microbial species exceeds the sum of biomass of what each species would produce on its own. Heavier biomass creates thicker, more protective barriers against cleaning agents and environmental stresses.
- Lines 215-218: Again, a repetition of the context described earlier.
We agree with this comment and have made the necessary changes
- I believe that Chapter 2.5 requires additional content.
The chapter was revised
- Table 2: I believe the column titled "Control measure" should be removed, as these issues appear later in the article.
We appreciate your suggestion. We consider the "Control measure" column necessary because it is directly linked to the specific cases described in the table.
- Tables 2, 3, 4, 5, and Figure 2: They lack any references to the literature from which they were based. Please provide additional information.
We agree with this comment and have made the necessary changes
- I expected more details after Chapter 4, which shouldn't be difficult to provide. Literature on this topic already exists, so please expand on this chapter. This will be crucial for food technologists, and providing a single example for each industry is insufficient.
We agree with this comment and extended Chapter 4
- Lines 315-327: Repeat about bacteriophages.
We agree with this comment and have made the necessary changes
- Lines 388-395: Please clarify whether this Listeria belongs to this group of raw materials/processed products, or if this is a more general comment.
We appreciate the reviewer’s attention to this detail. The mention of Listeria monocytogenes in lines 388–395 refers specifically to its relevance in the context of meat processing environments, as discussed in Section 4.2 of the manuscript. Our intention was to highlight its significance as a persistent contaminant in this specific sector, supported by outbreak data (Table 2).
- Lines 395-399: This is the third time I've mentioned planktonic forms, which are less resistant than biofilms :-(
We appreciate the reviewer’s observation. We have revised the manuscript to eliminate repeating ideas.
- Line 405: Italics are required for Latin names of microorganisms.
We agree with this comment and have made the necessary changes
- Line 487: Please provide information in which country the "Partnership for Food Safety Education's 'Fight BAC!'" operates.
The partnership operates in the U.S., the information was also added in the text.
- Line 493: What is the significance of "being aware of food recalls" for consumers? I can't connect this with biofilms and their importance to human health, because that's what was discussed in this subsection.
Being aware of food recalls is significant for consumers because biofilm-associated pathogens, such as Listeria monocytogenes or Salmonella spp., are frequent causes of contamination events leading to product recalls. These recalls serve as a critical public health measure to prevent exposure to persistent, biofilm-forming pathogens that may evade routine cleaning in processing environments
- Cited literature: Please ensure that all cited literature is recorded in a uniform manner. They should comply with the journal's requirements. Currently, there is a certain amount of discretion.
We agree with this comment and have made the requested changes.

Reviewer 3 Report
Comments and Suggestions for Authors
Thank you very much for the opportunity to read the review article entitled ‘A Comprehensive Review of Biofilm Formation, Control, and Economic Implications in Food Processing Environments.’
The article is written in a format appropriate for a review. The content of the paper is consistent with the title. The Authors have discussed in an interesting way both the mechanisms of biofilm formation, the microflora of biofilms and its interactions with the surrounding medium. The review of methods for determining biofilms and the influence of various factors on biofilms is very valuable. The examples of biofilm formation and their effects in various branches of the food industry were very valuable for food industry technologists. It is also very important that the Authors emphasise the economic losses associated with biofilms. The review ends with conclusions that are relevant to the content. The selection of literature is correct and sufficient.
While reading, I noticed a few imperfections that should be removed before publication:
1. I kindly ask the Authors to provide the numbers of the literature references for specific data in the tables. Similarly, it is necessary to add the numbers of the literature references under the titles of the figures.
2. In several places, I noticed that the names of microorganisms were not highlighted in italics (line 171, Table 3, line 405).
3. The Authors generally followed the rule of explaining abbreviations where they first appeared, but this was not the case with the abbreviation EPS (which first appears in line 84). Similarly, it would be worth placing the combination "free-floating (planctonic)" in line 53 instead of 80.
4. The sentence in lines 51-52 is largely duplicated by the sentence in lines 182-183.
5. If possible, please explain the term Agr QS in line 126.
6. Please also standardise the colour in the Author Contributions description
Apart from these minor editorial comments, I have no further comments and believe that, after the suggested minor corrections, this review article is highly valuable for publication.
Author Response
Reviewer 3
We sincerely thank the reviewer for the constructive comments and insightful suggestions, which have greatly contributed to improving the clarity, accuracy, and overall quality of our manuscript. We have addressed all points carefully and revised the text accordingly.
- I kindly ask the Authors to provide the numbers of the literature references for specific data in the tables. Similarly, it is necessary to add the numbers of the literature references under the titles of the figures.
We thank the reviewer for the valuable suggestion. We have revised the manuscript to include the corresponding reference numbers for all specific data points presented in the tables and figures.
- In several places, I noticed that the names of microorganisms were not highlighted in italics (line 171, Table 3, line 405).
We have carefully reviewed the manuscript and corrected the microbial names that were not properly italicized.
- The Authors generally followed the rule of explaining abbreviations where they first appeared, but this was not the case with the abbreviation EPS (which first appears in line 84). Similarly, it would be worth placing the combination "free-floating (planctonic)" in line 53 instead of 80.
We agree with this comment and have made the corresponding changes
- The sentence in lines 51-52 is largely duplicated by the sentence in lines 182-183.
We agree with this comment and have removed the repeating sentence.
- If possible, please explain the term Agr QS in line 126.
We agree with this comment and have made the corresponding explanation
- Please also standardise the colour in the Author Contributions description
We agree with this comment and have made the corresponding changes

Round 2
Reviewer 1 Report
Comments and Suggestions for Authors
accept
Reviewer 2 Report
Comments and Suggestions for Authors
I would like to thank the Authors for taking my comments into account and answering my questions.